# A comprehensive evaluation of cardiac amyloidosis epidemiology and diagnostics in French Guiana

Baptiste Desjardins[1], Kénol Franck[1], Nathalie Deschamps[1], Jean-Franky Alexis[1], Cyrille Mathien[1], Franck Boteko, Grace à Dieu Yabeta[2], Thibaud Damy[3], Jocelyn Inamo[4], Narcisse Elenga[1]*

**1** Centre Hispitalier de Cayenne, Avenue Alexis Baise, Cayenne, Guyane, France, **2** Centre Hospitaoier de l'Oues-Guyanais Franck Joly, Boulevard de la Liberté B, Saint-Laurent-du-Maroni, Guyane, France, **3** Hôpital Henri-Mondor A.P.-H.P, Rue Gustave Eiffel, Créteil, France, **4** Centre Hospitalier Universitaire de Martinique, Fort-de-France Cedex, Martinique, France

* elengafr@yahoo.fr

## Abstract

### Background

Cardiac amyloidosis (CA) is a potentially fatal systemic disease that has received increasing attention in recent years. However, there is no data on its epidemiology in French Guiana. This study aimed to evaluate the epidemiological characteristics of cardiac amyloidosis and describe the regional diagnostic pathways in French Guiana.

### Methods

We performed a multicenter retrospective study of Guianese patients with confirmed or suspected cardiac amyloidosis who were followed up in hospitals in French Guiana by private cardiologists.

### Results

A total of 47 patients were included. The study population was predominantly male (n=29, 61.7%). The mean age of the population was 72.8 years (SD=12.2). Most patients were from the island of Cayenne (n=34, 72.3%), had at least one cardiovascular risk factor (n=32, 68.1%), and more than half had extracardiac amyloid involvement (n=25, 53.2%). More than half of the patients were hospitalized at a reference center outside French Guiana (29, 61.7%), mainly at Henri Mondor Hospital (n=20, 69%) and Martinique (n=6, 20.7%). Bone scintigraphy was performed in 27 patients (57.5%) and hyperfixation was observed in 25 patients (93%). Anatomopathological examinations were performed in 33 patients (70.2%). Amyloid typing of the biopsied tissue revealed predominantly ATTR (n=14, 62.6%), AL amyloidosis (n=1, 4.5%), and AA amyloidosis (4.5%). Among the ATTR amyloidoses, we found mainly ATTRm

**Data availability statement:** All relevant data are within the manuscript and its Supporting Information files.

**Funding:** The author(s) received no specific funding for this work.

**Competing interests:** The authors have declared that no competing interests exist.

**Abbreviations:** AL, Amyloid light chain; SAA, Serum amyloid A; TTR, Transthyretin; CA, Cardiac amyloidosis; ECG, Electrocardiogram; BNP, B-type natriuretic peptide; PNDS, National Diagnostic and Treatment Protocols; EMB, Endomyocardial Biopsy; ATTR, Transthyretin amyloidosis; CHC, Centre Hospitalier de Cayenne; CHOG, Centre Hospitalier de l'Ouest Guyanais Franck Joly; CHK, Centre Hospitalier de Kourou; CHUM, Centre Hospitalier Universitaire de la Martinique; ICD-10, International Classification of Diseases-10th edition; MRI, magnetic resonance imaging; NYHA, New York Heart Association; LVEF, Left ventricular ejection fraction; IVS, Interventricular septum; ICD, Implantable cardioverter defibrillator; CVRF, Cardiovascular risk factor; RNA, Ribonucleic acid; AMM, Marketing authorization; siRNA, Small interfering RNA; BMI, Body mass index; CI, Confidence interval.

(n = 22, 75.8%). Genetic mutation testing was performed in approximately half of the patients (n = 25, 51.1%), mostly for the VAL122ILE mutation (n = 21, 84%), and in one case for the IL107VAL mutation (4%). Of the patients with ATTR amyloidosis, 22 (75.9%) were treated with tafamidis. Of the included patients, 18 (38.3%) died. The median overall survival (OS) was 38 months. Survival analysis from the date of diagnosis showed a probability of survival at 30 days, one year, 1.5 years and 4 years of 97% (95% confidence interval [CI]: 90–100), 68% (95%CI 55–84)), 64% (95%CI 51–79)), 32% (95%CI 29–41)), respectively.

## Conclusion

This study provides the first information on the diagnostic pathway for cardiac amyloidosis in French Guiana. The increasing proportion of undiagnosed patients has led us to create a French Guianese Amyloidosis Team to simplify the diagnostic pathway by focusing on cardiac MRI and biopsy, which can be performed locally. This is particularly important, as current and future therapeutic advances mean that more effective treatments are on the horizon.

## Introduction

Amyloidosis is a rare systemic disease characterized by the deposition of abnormally folded proteins in the extracellular space in a beta-pleated-sheet conformation. Amyloid substances are composed of two groups of proteins: a common structure (consisting mainly of amyloid component P, proteoglycans, protease inhibitors, and apolipoprotein E) and a protein specific to the type of amyloidosis, which forms the basis of the biochemical classification of amyloidosis [1]. The main types of amyloidosis include immunoglobulin light chain amyloidosis (AL amyloidosis), amyloid A (AA) amyloidosis and transthyretin (TTR) amyloidosis (hereditary or familial amyloidosis) formed from transthyretin [1]. To date, approximately 42 amyloidogenic precursor proteins have been identified, of which 9 appear to be capable of accumulating in cardiac extracellular tissue. The extracellular deposition of misfolded proteins in cardiac interstitial tissue between myocardial fibres leads to cardiac amyloidosis (CA), which is often overlooked as a cause of diastolic heart failure [2]. Heart involvement can occur as part of a systemic disease or as a localised phenomenon. The diagnostic approach for suspected cardiac amyloidosis is defined in the 2021 National Diagnostic and Treatment Protocol (PNDS) for CA [3, 4]. More than 120 mutations have been identified to date. Depending on the type of mutation, the organs affected and clinical manifestations of the disease differ. The prevalence of different types of mutations also varies among ethnic groups and geographic areas. For example, the Val122Ile mutation is more common in Africans, Afro-Americans, and Afro-Caribbeans, and cardiac involvement is more prominent than neuropathic involvement in these patients [5]. In the last two decades, advances in cardiac imaging and approval of disease-modifying drugs for ATTR and AL have contributed significantly to the increased recognition of cardiac amyloidosis in routine practice [6, 7]. While the

median survival without treatment is estimated to be 2.5 years for ATTRm and 3.5 years for ATTRwt in untreated patients, an ATTR-stabilizing treatment, tafamidis, has been available in many countries since 2018 [8]. Most studies are conducted in high-income countries with important patient populations [9–11]. There are few studies on CA in the Latin-American and Caribean countries. However, the results of a recent survey demonstrate a significant proportion of health professionals dealing with scarce information about CA in the Latin-American region [12]. French Guiana is an overseas French territory located northeast of the South American continent. The population is historically multi-ethnic and multicultural, with a majority of people of African descent [13]. There have been no studies on cardiac amyloidosis in French Guiana, although the disease is becoming better known. Limited knowledge of the current epidemiology of cardiac amyloidosis in Latin American and Caribbean countries is a major obstacle to the effective management of this disease. For this reason, we conducted a retrospective study using existing data to perform an epidemiological analysis of cardiac amyloidosis in French Guiana using a population-based approach and to describe the care pathway.

## Methods

### Type of study

This descriptive, multicenter, retrospective study involving three public hospitals and private cardiologists in French Guiana.

### Study population

**Inclusion criteria.** Patients with or suspected of having amyloidosis of any type (AL, AA, TTR) with cardiac involvement, residing in French Guiana, and having access to healthcare between 2002 and 2022 were included in this study.

**Exclusion criteria.** The exclusion criteria were as follows: age less than 18 years and patients not resident in French Guiana.

**Data collection method.** The identification of affected patients in hospital centers was based on stays coded for amyloidosis (International Classification of Diseases (ICD-1O) code E85x). In the case of CHC, data collection began in 2007 because of the availability of patient records in the CHC archiving software (Cora). Patients followed up by private cardiologists were interviewed. To be as comprehensive as possible, we used all available data from hospitals and private cardiologists, taking into account that the majority of patients were only followed up in the hospital and that all private practice patients went to the hospital for their check-up and final diagnosis. In addition, all cardiologists in French GuIana work on a network and share the same equipment suppliers and protocols, which are based on those of the French National Health Authority. Both private and hospital cardiologists collected data from patients suspected of having amyloidosis, as protocolized.

### Diagnosis of amyloidosis

Amyloidosis was diagnosed based on a pathological biopsy with a positive Congo red test or a positive bone scan with a Perrugini score of 2 or 3.

Amyloidosis was suspected in cases of high clinical suspicion with compatible magnetic resonance imaging (MRI) and/or echocardiographic images in patients in whom bone scintigraphy and pathological examination could not be obtained.

### Data collected

Demographic data (age, sex, geographic origin, weight, and height), medical data (New York Heart Association (NYHA) dyspnea, extracardiac damage (carpal tunnel, narrow lumbar canal, Dupuytren's disease, peripheral neuropathy, deafness, hypotension, and dysautonomia), cardiac biomarkers, echocardiographic data (left ventricular ejection fraction (LVEF), interventricular septum (IVS), longitudinal strain), cardiac scintigraphy (Perrugini score and heart/mediastinum ratio), pathology (Congo red staining and typing), search for genetic mutations, treatment of heart failure, and specific treatment of amyloidosis (tafamidis)).

Clinical data were extracted from hospital admission and consultation reports using the Cora Document software at the CHC. Biological data were extracted using the CyberLab software. Imaging data were extracted using Xplore software. At CHK and CHOG, all patient data were extracted from paper archives.

The records of patients followed by private cardiologists were retrieved using their respective softwares.

These records were collected at the time of amyloidosis diagnosis. For patients in whom amyloidosis was suspected but was not confirmed, records were collected at the time of the first suspicion of amyloidosis.

We also collected the date of first suspicion of amyloidosis and, for these patients, the dates of visits to the reference centers. We were able to visit these reference centers to complete the clinical data of these patients (Henri Mondor Hospital in Créteil and the University Hospital Centre of Martinique). Data were collected from January to June 2024, and statistical analyses were carried out from June to September 2024.

### Statistical methods

The collected data were entered into an Excel database for the analysis. We checked our data for normality before applying the appropriate statistical tests. Patient characteristics were described according to their type by median and quartiles or numbers and percentages, and compared according to the variables of interest using parametric and non-parametric tests. For categorical variables, the parametric chi2 test was used; otherwise, the nonparametric Fisher's exact test was used. For continuous variables, the parametric Student's t-test was used; otherwise, the non-parametric Wilcoxon test was used. The Kaplan-Meier method was used to estimate survival with event = death and the probability of non-diagnosis with event = diagnosis. Statistical significance was set at $P < 0.05$. Data were analyzed using R statistical software version 4.3.1. Regression imputation was used to impute missing data.

### Regulatory and ethical issues

This study was part of non-human subject research and falls under the "Reference Methodology" (MR-004), for which the CHC signed a compliance commitment dated September 25, 2002 (see supporting information 1, 2, 3 and 4). A privacy impact assessment was conducted, and a summary of the study was published on the Health-Data-Hub website (https://health-data-hub.fr/projets/etude-epidemiologique-descriptive-des-patients-atteins-damylose-aa-avec-atteinte-cardiaque, N° F20231026201407). The legal basis for processing data is the mission of the public interest. The study was registered in the data processing register by the data protection officer of the Cayenne Hospital. Informed consent was obtained by mail, and all patients received an information sheet and non-objection form. Failure to return the form one month after it was sent was considered consent. During or after data collection or access, we had no access to information that could identify the individual participants.

### Study results

Using hospital databases, we compiled an initial list of 70 cases of confirmed or suspected cardiac amyloidosis. We included 19 patients followed by the only private cardiologist in Cayenne, resulting in a total of 89 patients with suspected or confirmed cardiac amyloidosis. Of these patients, 37 were excluded for various reasons: no cardiac involvement, patients under 18 years of age, patients not living in French Guiana, and coding errors (patients identified by hospital coding as E85 but with amyloid neuropathy only). Four patients refused to participate in this study. (Fig 1).

### Patient characteristics

The patient characteristics are shown in Table 1.

The study population was predominantly male (n = 29, 61.7%). The mean age of the population was 72.8 years (SD = 12.2). The majority of the patients were from the island of Cayenne (n = 34, 72.3%). The remaining patients were from Kourou (n = 8, 17%), Saint-Laurent du Maroni (n = 4, 8.5%), and Saint Georges (n = 1, 2.1%).

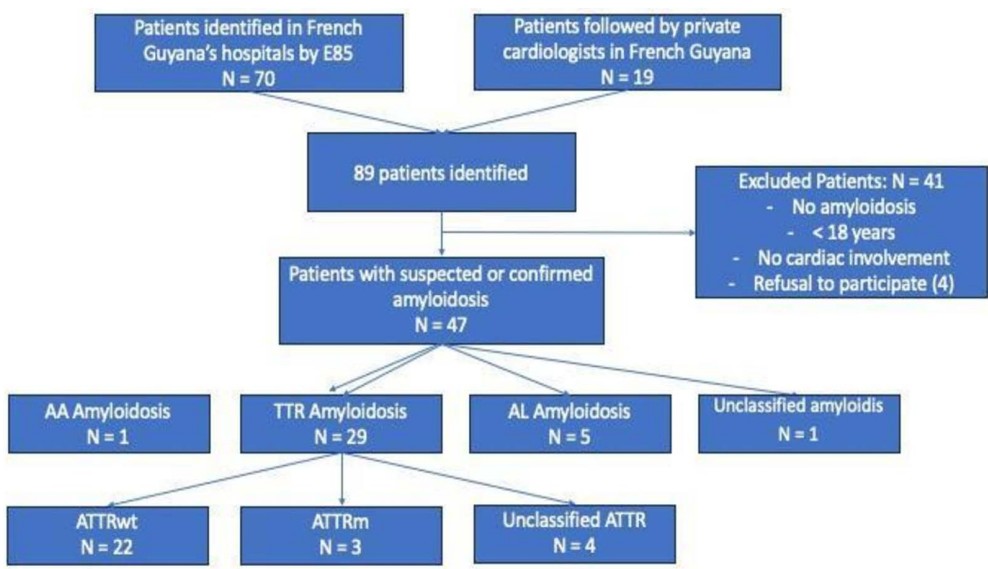

**Fig 1. Flow-chart of patient selection.**

At the time of diagnosis, the mean weight was 72 kg (sd = 16.3) and the mean height was 170.9 cm (sd = 9.7), with a mean body mass index (BMI) of 24.5 kg/m2 (sd = 5.2).

Most patients had at least one cardiovascular risk factor (n = 32, 68.1%), hypertension (n = 32, 68.1%), diabetes (n = 7, 14.9%), and dyslipidemia (n = 4, 8.5%).

Cardiac involvement was assessed at the time of diagnosis (or at the time of first suspicion in patients without a confirmed diagnosis) using the NYHA scale, mainly stages II (n = 16, 34%) and III (n = 12, 25.5%), and the presence or absence of microvoltage on ECG, which was found in 15 patients (31.9%). Echocardiographic impairment was assessed by LVEF, interventricular septum measurement, and longitudinal strain, and the mean LVEF was 49.7% (SD = 16) and less than 50% in 44.7% of the patients (n = 21); IVS was greater than 12 mm in 66% of the patients (n = 31). The mean longitudinal strain measurement was −9.3% (available for only 24 patients).

Cardiac markers (troponin and N-terminal (NT) pro-BNP) were elevated in all patients, with a mean troponin level of 85.9 ng/ml (sd = 55.4) and a mean NT pro-BNP level of 6543.1 pg/ml (sd = 8231.8). BNP of 6543.1 pg/ml (sd = 8231.8).

More than half of the patients presented with extracardiac involvement in amyloidosis (n = 25, 53.2%). The most common extracardiac manifestations were, in order, carpal tunnel syndrome (n = 18, 38.3%), peripheral neuropathies (n = 11, 23.4%), dysautonomia (n = 7, 14, 9%), orthostatic hypotension (n = 6, 12.8%), and deafness (n = 5, 10.6%). Only one patient had Dupuytren's disease (2.1%).

### Diagnostic approach

More than half of the patients were hospitalized at an amyloidosis reference center outside the territory of Guyana (29, 61.7%), mainly at Henri Mondor Hospital (n = 20, 69%) and CHUM (n = 6, 20.7%).

Cardiac magnetic resonance imaging (MRI) was performed in 29 patients (61.7%) to detect amyloidosis.

Bone scintigraphy, the reference test for diagnosis, was performed in 27 patients (57.5%) and hyperuptake was observed in 25 patients (93%).

Anatomopathological examinations were performed in 33 patients (70.2%). The type of biopsy performed was predominantly salivary gland (n = 25, 53.2%). Congo red staining was positive in 25 (92.6%) patients. Amyloidosis typing of the

**Table 1. Patient characteristics.**

| Variable | All patients (n=47) | ATTR (n=29) |
|---|---|---|
| Age at diagnosis | 72,8 (sd=12,2) | 75.4 (sd=7.9) |
| Male sex | 29 (61,7%) | 17 (58.6%) |
| Weight | 72 (sd=16,3) | 71.6 (sd=13.1) |
| Size | 170,9 (sd=9,7) | 171.8 (sd=10.3) |
| BMI | 24,5 (sd=5,2) | 24.3 (sd=4.5) |
| **Geographical origin** | | |
| Cayenne | 34 (72,3%) | 22 (75.9%) |
| Eastern Guyana | 1 (2,1%) | 1 (3.4%) |
| Kourou | 8 (17%) | 5 (17.2%) |
| West Guiana | 4 (8,5%) | 1 (3.4%) |
| **Case history** | | |
| At least 1 CVRF | 32 (68,1%) | 21 (72.4%) |
| Arterial hypertension | 32 (68,1%) | 21 (72.4%) |
| Diabetes | 7 (14,9%) | 4 (13.8%) |
| Dyslipidemia | 4 (8,5%) | 4 (13.8%) |
| Tobacco | 0 (0%) | 0 (0%) |
| Alcohol | 1 (2,1%) | 1 (3.4%) |
| **Clinical damage** | | |
| Systolic blood pressure | 130 (sd=24) | 128.3 (sd=23.6) |
| Diastolic blood pressure | 76 (sd=14,7) | 75.1 (sd=14.7) |
| Heart rate | 78,7 (sd=12,2) | 76.9 (sd=10.2) |
| *NYHA* | | |
| I | 7 (14,9%) | 3 (10.3%) |
| II | 16 (34%) | 13 (44.8%) |
| III | 12 (25,5%) | 8 (27.6%) |
| IV | 5 (10,6%) | 3 (10.3%) |
| Previous myocardial infarction | 1 (2,1%) | 0 (0%) |
| *ECG* | | |
| Microvoltage | 15 (31,9%) | 7 (26.9%) |
| Sinusal | 29 (61,7%) | 7 (25%) |
| Atrial fibrillation | 10 (21,3%) | 18 (64.3%) |
| Stimulation | 4 (8,5%) | 3 (10.7%) |
| *Biology* | | |
| Troponin | 85,9 (sd=55,4) | 85.9 (sd=55.4) |
| NT pro BNP | 6543,1 (sd=8231,8) | 6080.4 (sd=7649.9) |
| Hemoglobin | 12,8 (sd=2) | 13.1 (sd=2.1) |
| Creatininemia | 122,8 (sd=87,7) | 108.2 (sd=35.4) |
| Gammapathy | 7 (14,9%) | 1 (3.4%) |
| *Echocardiographic criteria* | | |
| LVEF | 49,7 (sd=16) | 46.5 (sd=15) |
| LVEF<50% | 20 (42.6%) | 15 (51.7%) |
| IVS | 17.7 (sd=3.5) | 17.5 (sd=3.7) |
| IVS>12 mm | 31 (66%) | 23 (79.3%) |
| Strain | −9.3 (sd=3.8) | −8.9 (sd=3.8) |
| *Extracardiac disorders* | 25 (53,2%) | 21 (72.4%) |
| Carpal tunnel syndrome | 18 (38,3%) | 17 (58.6%) |

*(Continued)*

**Table 1.** (Continued)

| Variable | All patients (n = 47) | ATTR (n = 29) |
|---|---|---|
| Narrow lumbar canal | 5 (10.6%) | 5 (17.2%) |
| Dupuytren | 1 (5.6%) | 1 (3.4%) |
| Peripheral neuropathy | 11 (23,4%) | 9 (31%) |
| Dysautonomia | 7 (14.9%) | 6 (20.7%) |
| Hypotension | 6 (12.8%) | 5 (17.2%) |
| Deafness | 5 (10.6%) | 5 (17.2%) |
| **Imaging tests** | | |
| MRI performed | 29 (61.7%) | 21 (72.4%) |
| *Bone scan* | | |
| Hyperfixation | 25 (53.2%) | 25 (86.2%) |
| Negative | 2 (4.3%) | 1 (3.4%) |
| Not carried out | 20 (42.6%) | 3 (10.3%) |
| Heart to mediastinum ratio | 1.4 (Q1;Q3=[1.3; 1.6]) | 1.4 (Q1;Q3=[1.3; 1.6]) |
| Perrugini 2 | 2 (4.3%) | 2 (6.9%) |
| Perrugini 3 | 3 (6.4%) | 3 (10.3%) |
| **Anatomopathology** | | |
| Not done | 13 (27.7%) | 3 (10.3%) |
| Salivary gland biopsy | 25 (53.2%) | 20 (69%) |
| Endocardium | 4 (8.5%) | 4 (13.8%) |
| Abdominal fat | 1 (2.1%) | 1 (3.4%) |
| kidney biopsy | 2 (4.3%) | 0 (0%) |
| Type of biopsy unknown | 1 (2.1%) | |
| *Results* | | |
| Positive | 27 (57,4%) | 21 (72.4%) |
| Congo red positive | 25 (53,2%) | 20 (69%) |
| *Typing* | | |
| TTR | 14 (29.8%) | 14 (48.3%) |
| AA | 1 (2.1%) | 0 (0%) |
| AL | 1 (2.1%) | 0 (0%) |
| Unknown | 6 (12.8%) | 5 (17.2%) |
| **Genetic analysis** | | |
| Realized | 25 (53.1%) | 25 (86.2%) |
| *Location* | | |
| CHUM | 4 (16%) | 4 (13.8%) |
| Henri Mondor Center | 21 (84%) | 21 (72.4%) |
| *Mutations* | | |
| VAL122ILE | 21 (44.7%) | 21 (72.4%) |
| ILE107VAL | 1 (2.1%) | 1 (3.4%) |
| **Diagnosis** | | |
| ATTR | 29 (61.7%) | 29 (100%) |
| ATTRm | 22 (75.9%) | 22 (75.9%) |
| ATTRwt | 3 (10.3%) | 3 (10.3%) |
| AA | 1 (2.1%) | |
| AL | 5 (10.6%) | |
| Untyped amyloidosis | 1 (2.1%) | |

*(Continued)*

**Table 1.** (Continued)

| Variable | All patients (n = 47) | ATTR (n = 29) |
|---|---|---|
| Suspected amyloidosis | 11 (23.4%) | |
| **Treatments** | | |
| Tafamidis | 22 (46.8%) | 22 (75.9%) |
| 20 mg | 6 (12.8%) | 6 (20.7%) |
| 61 mg | 14 (29.8%) | 14 (48.3%) |
| 80 mg | 1 (2.1%) | 1 (3.4%) |
| Diuretic | 32 (68.1%) | 20 (69%) |
| Beta-blocker | 1 (2.1%) | 0 (0%) |
| Pacemaker | 8 (17%) | 7 (24.1%) |
| ICD | 9 (19.1%) | 9 (31%) |
| Chemotherapy | 6 (12.8%) | 0 (0%) |
| **Reference center** | | |
| CHUM | 6 (12.8%) | 5 (17.2%) |
| Henri Mondor Center | 20 (42.6%) | 20 (69%) |
| Other | 3 (6,4%) | 1 (3.4%) |
| No | 18 (38.3%) | 3 (10.3%) |
| **Deaths** | 18 (38.3%) | 12 (41.4%) |
| *Cause of death* | | |
| Cardiopulmonary arrest | 1 (5.6%) | |
| Cardiogenic shock | 1 (5.6%) | 1 (8.3%) |
| Multivisceral failure | 3 (16.7%) | 0 (0%) |
| End-stage heart failure | 5 (27.8%) | 5 (41.7%) |
| Infection | 2 (11.1%) | 2 (16.7%) |
| Extracardiac | 1 (5.6%) | 1 (8.3%) |
| Unknown | 5 (27.8%) | 3 (25%) |

biopsied tissue was predominantly ATTR (n = 14, 62.6%), AL amyloidosis (n = 1, 4.5%), and AA amyloidosis (4.5%). The type was unknown in 27.3% of the cases (n = 6).

Genetic mutation testing was performed in approximately half of the patients (n = 25, 51.1%). It was performed at the reference centers for all patients, mainly at the Center Hospitalier Henri Mondor (n = 21, 84%), and the rest at the CHUM (n = 4, 16%). Genetic testing revealed a VAL122ILE mutation in the vast majority of cases (n = 21, 84%) and an IL107VAL mutation in one case (n = 1, 4%). Three patients (12%).

Of the 47 patients, 11 did not have a confirmed diagnosis of amyloidosis but only a suspected diagnosis (23.4%). In the remaining patients, the different types of amyloidosis were mainly ATTR (n = 29, 61.7%), followed by AL amyloidosis (n = 5, 10.6%), and AA (n = 1, 2.1%). One patient had an atypical amyloidosis.

## ATTR patients

Among ATTR amyloidoses, we found mainly ATTRm (n = 22, 75.8%), followed by ATTRwt (n = 3, 10.3%). The type of ATTR was unknown in four patients (13.8%). The mean age was 75.9 years (SD = 7.9), with a male predominance (n = 17, 58.6%) and a mean BMI of 24.3 kg/m2 (st = 4.5).

Cardiac involvement was mainly NYHA II dyspnea (13 cases, 48.1%), with a mean LVEF of 46.5% (SD = 15) and elevated cardiac biomarker levels.

The majority of these patients presented with extracardiac involvement (n = 21, 72.4%), mainly carpal tunnel syndrome (n = 17, 58.6%), peripheral neuropathy (n = 9, 31%), dysautonomia (n = 6, 20.7%), narrow lumbar canal (n = 5, 17.2%), hypotension (n = 5, 17.2%), and deafness (n = 5, 17.2%) (2%).

## Treatments

Of the patients with ATTR amyloidosis, 22 received amyloidosis-specific treatment with tafamidis (75.9%). The majority received a dose of 61 mg/day (n = 14, 66.7%), while the others received only 20 mg/day (n = 6, 46.8%) or 80 mg/day (n = 1, 4.8%). Nine patients had an implantable cardioverter defibrillator (ICD) (19.1%) and eight patients had a pacemaker (17%).

One patient had a combined heart-liver transplant.

Regarding symptomatic treatment of heart failure, 32 patients were treated with a diuretic (68.1%) and one patient was treated with a beta-blocker (2.1%).

Only 5 patients with AL were identified. The 6th was receiving chemotherapy for conditions other than AL.

## Comparisons by subgroups

**Comparison by gender.** Men and women were compared (Table 2). Statistically significant differences were found in cardiovascular risk factors (CVRF) and echocardiographic measurements.

The absence of CV RF was significantly higher in the male group (n = 14 [48.3%] vs. 1 [5.6%], p = 0.031), with a much higher proportion of women being followed up for hypertension (17 cases, 94.4% vs. 14, 48.3%, p = 0.031).

Regarding echocardiographic data, the LVEF was significantly higher in women than in men (59 [50, 69.5]% vs. 40 [35.2; 54.8]%, p = 0.0023), and the longitudinal strain was significantly lower in women (−11 [−13.5; −10]% vs. −8.2 [−9.9; −5.4]%, p = 0.0428).

*Comparison of the confirmed amyloidosis group (ATTR, AL, AA, untyped amyloidosis) with the suspected amyloidosis group* (Fig 2).

When comparing patients with a confirmed diagnosis of amyloidosis to those with suspected amyloidosis, the only statistically significant difference was extracardiac involvement of amyloidosis (25 cases, 69.4% vs. 0.0%, p < 0.001). Carpal tunnel syndrome (18 patients, 50% vs. 0.0%, p = 0.032) and peripheral neuropathy (11 patients, 30.6% vs. 0.0%, p = 0.0457) were also significantly different between the two groups.

## Survival

Of the patients enrolled in this study, 18 (38.3%) died. The causes of death were terminal heart failure (5 cases, 27.8%), multiple organ failure (3 cases, 16.7%), infection (2 cases, 11.8%), cardiorespiratory arrest (1 case, 5.6%), and cardiogenic shock (1 case, 5.6%). The cause of death was unknown in five patients (27.8%). Table 3 summarizes the characteristics of the patients by diagnosis and shows that the number of diagnoses confirmed at the reference center was higher (p < 0.001).

**Survival analyses.** Survival analyses were performed using the Kaplan-Meier method to evaluate the time from the first suspected diagnosis to the actual diagnosis and survival from the date of diagnosis. In this curve (Fig 3), we consider the event to be the diagnosis; therefore, the curve corresponds to the probability of nondiagnosis. The probability of diagnosis was 6% at 30 days, 66% at one year, 73% at 1.5 years and 87% at 3.5 years.

The median overall survival (OS) was 38 months. Survival analysis from the date of diagnosis showed a probability of survival at 30 days, one year, 1.5 years and 4 years of 97% (95% confidence interval [CI]: 90–100), 68% (95%CI 55–84)), 64% (95%CI 51–79)), 32% (95%CI 29–41)), respectively (Fig 4).

## Discussion

This study represents the first epidemiological analysis of patients with cardiac amyloidosis in French Guiana. Little information is available about patients with cardiac amyloidosis in Latin American countries close to French Guiana (especially

**Table 2. Comparison of female vs male characteristics.**

| Variable | Female (n = 18) | Male (n = 29) | p.value |
|---|---|---|---|
| Age at diagnosis | 77 q1;q3=[71.9;82.1] | 74.8 q1;q3=[66.6;77.2] | 0.069 |
| Weight | 62 q1;q3=[54;70] | 74.5 q1;q3=[69;80] | 0.1088 |
| Size | 160 q1;q3=[157.5;165] | 174 q1;q3=[171;179.8] | <0.001 |
| BMI | 22.9 q1;q3=[19;25.9] | 24 q1;q3=[22.2;25.8] | 0.3396 |
| **Geographical origin** | | | |
| Cayenne | 13 (72.2%) | 21 (72.4%) | 0.1369 |
| Eastern Guyana | 0 (0%) | 1 (3.4%) | |
| Kourou | 5 (27.8%) | 3 (10.3%) | |
| West Guiana | 0 (0%) | 4 (13.8%) | |
| **Case history** | | | |
| At least 1 CVRF | 17 (94.4%) | 15 (51.7%) | 0.0031 |
| Arterial hypertension | 17 (94.4%) | 15 (51.7%) | 0.0031 |
| Diabetes | 5 (27.8%) | 2 (6.9%) | 0.0892 |
| Dyslipidemia | 3 (16.7%) | 1 (3.4%) | 0,1498 |
| Tobacco | 18 (100%) | 29 (100%) | |
| Alcohol | 0 (0%) | 1 (3.4%) | 1 |
| **Clinical damage** | | | |
| Systolic blood pressure | 138 q1;q3=[125.5;148.5] | 125.5 q1;q3=[111.2;138] | 0.092 |
| Diastolic blood pressure | 69.5 q1;q3=[65.2;76.5] | 75.5 q1;q3=[70;86.2] | 0.3125 |
| Heart rate | 75 q1;q3=[66.8;86.8] | 78 q1;q3=[71.5;87] | 0.6704 |
| *NYHA* | | | |
| I | 3 (18.8%) | 4 (16.7%) | 0.907 |
| II | 7 (43.8%) | 9 (37.5%) | |
| III | 5 (31.2%) | 7 (29.2%) | |
| IV | 1 (6.2%) | 4 (16.7%) | |
| Previous MI | 0 (0%) | 1 (3.4%) | 1 |
| *ECG* | | | |
| Microvoltage | 3 (17.6%) | 12 (46.2%) | 0.1005 |
| Sinusal | 8 (50%) | 21 (77.8%) | |
| Atrial fibrillation | 7 (43.8%) | 3 (11.1%) | 0.0405 |
| Stimulation | 1 (6.2%) | 3 (11.1%) | |
| *Biology* | | | |
| Troponin | 52 q1;q3=[49;70] | 76.9 q1;q3=[64.2;129.5] | 0.0507 |
| NT pro BNP | 3355.5 q1;q3=[1408.5;6898.8] | 4727 q1;q3=[1772.5;7856] | 0.6955 |
| Hemoglobin | 11 q1;q3=[10.3;12.9] | 13.1 q1;q3=[12.2;14.9] | 0.0316 |
| Creatininemia | 99 q1;q3=[81.5;126] | 110 q1;q3=[86;137] | 0.5295 |
| Gammapathy | 2 (13.3%) | 5 (20%) | 0.6913 |
| *Echocardiographic criteria* | | | |
| LVEF | 59 q1;q3=[50;69.5] | 40 q1;q3=[35.2;54.8] | 0.0023 |
| LVEF < 50% | 4 (26.7%) | 16 (61.5%) | 0.0516 |
| IVS | 17 q1;q3=[13.5;19] | 18 q1;q3=[16;20.2] | 0.1289 |
| IVS > 12 mm | 8 (72.7%) | 23 (95.8%) | 0.0819 |
| Strain | −11 q1;q3=[−13.5;-10] | −8.3 q1;q3=[−9.9;-5.4] | 0.0428 |
| *Extracardiac damage* | 9 (50%) | 16 (55.2%) | 0.9643 |
| Carpal tunnel syndrome | 8 (44.4%) | 10 (34.5%) | 0,7082 |
| Narrow lumbar canal | 1 (5.6%) | 4 (13.8%) | 0,6355 |

*(Continued)*

**Table 2.** (Continued)

| Variable | Female (n = 18) | Male (n = 29) | p.value |
|---|---|---|---|
| Dupuytren | 1 (5.6%) | 0 (0%) | 0.383 |
| Peripheral neuropathy | 3 (16.7%) | 8 (27.6%) | 0.4921 |
| Dysautonomia | 2 (11.1%) | 5 (17.2%) | 0.6918 |
| Hypotension | 3 (16.7%) | 3 (10.3%) | 0.6616 |
| Deafness | 3 (16.7%) | 2 (6.9%) | 0.3568 |
| **Imaging tests** | | | |
| MRI performed | 11 (61.1%) | 18 (69.2%) | 0.814 |
| *Bone scan* | | | |
| Hyperfixation | 9 (50%) | 16 (55.2%) | 0.4503 |
| Negative | 0 (0%) | 2 (6.9%) | |
| Not carried out | 9 (50%) | 11 (37.9%) | |
| Heart to mediastinum ratio | 1.6 q1;q3=[1.3;1.9] | 1.3 q1;q3=[1.3;1.4] | 0.4113 |
| Perrugini 2 | 0 (0%) | 2 (50%) | 1 |
| Perrugini 3 | 1 (100%) | 2 (50%) | |
| **Anatomopathology** | | | |
| Not done | 7 (38.9%) | 6 (21.4%) | |
| Salivary gland biopsy | 9 (50%) | 16 (57.1%) | |
| Endocardium | 1 (5.6%) | 3 (10.7%) | |
| Abdominal fat | 0 (0%) | 1 (3.6%) | |
| kidney biopsy | 1 (5.6%) | 1 (3.6%) | |
| Type of biopsy unknown | 0 (0%) | 1 (3.6%) | |
| *Results* | | | |
| Positive | 8 (80%) | 19 (86.4%) | 0.6367 |
| Congo red positive | 8 (100%) | 17 (89.5%) | 1 |
| *Typing* | | | |
| TTR | 5 (71.4%) | 9 (60%) | |
| AA | 1 (14.3%) | 0 (0%) | 0.4158 |
| AL | 0 (0%) | 1 (6.7%) | |
| Unknown | 1 (14.3%) | 5 (33.3%) | |
| **Genetic analysis** | | | |
| Realized | 7 (38.9%) | 18 (62%) | 0.3099 |
| *Location* | | | |
| CHUM | 0 (0%) | 4 (23.5%) | 0.2689 |
| Henri Mondor Center | 8 (100%) | 13 (76.5%) | |
| *Mutations* | | | |
| VAL122ILE | 7 (87.5%) | 14 (87.5%) | |
| ILE107VAL | 0 (0%) | 1 (6.2%) | 1 |
| **Diagnosis** | | | |
| ATTR | 12 (66.7%) | 17 (58.6%) | |
| ATTRm | 7 (58.3%) | 15 (88.2%) | 0.2079 |
| ATTRwt | 2 (16.7%) | 1 (5.9%) | |
| AA | 1 (5.6%) | 0 (0%) | 0.7026 |
| AL | 1 (5.6%) | 4 (13.8%) | |
| Untyped amyloidosis | 0 (0%) | 1 (3.4%) | |
| Suspected amyloidosis | 4 (22.2%) | 7 (24.1%) | |

*(Continued)*

**Table 2.** (Continued)

| Variable | Female (n = 18) | Male (n = 29) | p.value |
|---|---|---|---|
| **Treatments** | | | |
| Tafamidis | 10 (55.6%) | 12 (41.4%) | 0.5182 |
| 20 mg | 4 (40%) | 2 (18.2%) | 0.4903 |
| 61 mg | 6 (60%) | 8 (72.7%) | |
| 80 mg | 0 (0%) | 1 (9.1%) | |
| Diuretic | 13 (72.2%) | 19 (70.4%) | 1 |
| Beta-blocker | 1 (5.6%) | 0 (0%) | 0.4 |
| Pacemaker | 2 (11.1%) | 6 (20.7%) | 0.6918 |
| ICD | 3 (16.7%) | 6 (20.7%) | 1 |
| Chemotherapy | 2 (11.1%) | 4 (13.8%) | 1 |
| **Reference center** | | | |
| CHUM | 1 (10%) | 5 (26.3%) | |
| Henri Mondor Center | 1 (10%) | 5 (26.3%) | |
| Other | 1 (10%) | 2 (10,6%) | |
| No | 8 (44.4%) | 10 (34.5%) | 0.7082 |
| **Deaths** | 8 (44.4%) | 10 (34.5%) | 0.7082 |
| *Cause of death* | | | |
| Cardiopulmonary arrest | 1 (12.5%) | 0 (0%) | |
| Cardiogenic shock | 0 (0%) | 1 (10%) | |
| Multivisceral failure | 2 (25%) | 1 (10%) | |
| End-stage heart failure | 2 (25%) | 3 (30%) | |
| Infection | 2 (25%) | 0 (0%) | |
| Extra-cardiac | 0 (0%) | 1 (10%) | |
| Unknown | 1 (12.5%) | 4 (40%) | |

Suriname and Guyana). This could be explained by the fact that cardiac amyloidosis is a rare disease that requires many diagnostic resources, which are often not available in developing countries [14]. However, our results can be compared with those from other countries in South America and the Caribbean. In Colombia, of 31 patients with amyloidosis, 17 had ATTR amyloidosis and 14 had AL amyloidosis. Overall mortality was 25%. The mean age at diagnosis was 74 years, with a male predominance. The most common comorbidities were hypertension and atrial fibrillation. The most common clinical presentation was congestive heart failure (75%) with mildly reduced ejection fraction (41.94%), followed by reduced ejection fraction (32.26%) and preserved ejection fraction (25.81%). In the ATTR subtype, 41.18% had a reduced ejection fraction and 35.29% had a mildly reduced ejection fraction [15] These results are in line with our. An increase in incidence and mortality was observed in Argentina [16]. The crude incidence rate was 63 (95% CI: 52–76) and the crude mortality rate was 31 (95% CI: 23–40) cases per million person-years, with the ATTRwt subtype showing the highest rates. Men were more affected, especially those over 70 years of age. We were not able to determine incidence rates because of the study design. In Brazil, a 2018 study described the characteristics of patients with amyloidosis [17]. They found a younger population, with a mean age of 66 years (compared to 72.8 years in our study), but a similar proportion of men (59%). The type of amyloidosis was also slightly different in this study, with 80% ATTR and 20% AL. However, this information is difficult to compare with the population in our study because of the high proportion of suspected rather than confirmed amyloidosis cases in our population. The lower proportion of AL amyloidosis in our study could be explained by the fact that patients with hematologic diseases are more quickly transferred to hematology centers in mainland France and, therefore, not followed up for their cardiac damage related to amyloidosis in French Guiana. The most frequently

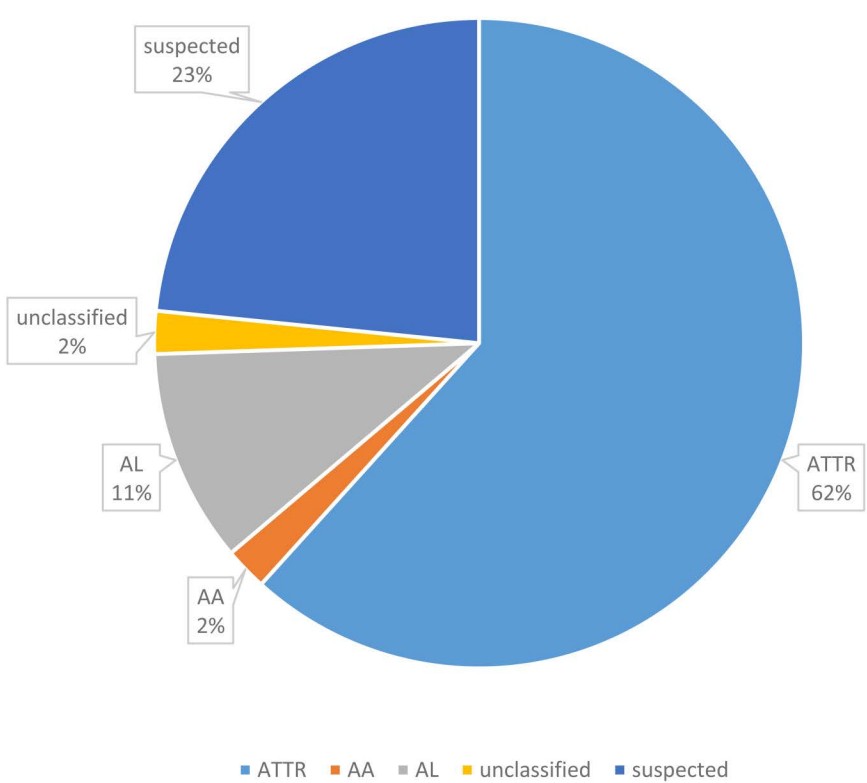

**Type of amyloidosis**

- ATTR 62%
- AA 2%
- AL 11%
- unclassified 2%
- suspected 23%

■ ATTR  ■ AA  ■ AL  ■ unclassified  ■ suspected

**Fig 2. Types of amyloidosis.**

associated mutations were also different, with some mutations in Brazil explained by the high proportion of Portuguese in the ethnic composition of the Brazilian population, and others by Afro-descendant populations [18]. Compared to these studies, the number of cases we reported is relatively small. It is therefore plausible to assume that there was a selection bias in our study, especially as the majority of patients came from Cayenne.

Recent studies in Latin America have demonstrated the need for educational programs to increase clinical awareness and the early detection of cardiac amyloidosis [12, 19].

With regard to ATTRm, the most common mutation found was Val122Ile, as in our population, which is also the case in the United States, where 3–4% of African Americans carry this mutation [20].

In the French West Indies, various studies carried out on patients with ATTR showed results comparable to those of our study, particularly with regard to the origin, age, and biometric characteristics of the patients [21]. The proportion of men was slightly higher (70%). However, these data only included patients with ATTR. No data were available for patients with any type of cardiac amyloidosis. The most common mutation in ATTRm in the Martinique population was Val122Ile. These similarities between the two populations in these studies can be explained by the similar origin of the populations, both African and slavery.

**Table 3. Patient characteristics according to diagnosis (confirmed vs. suspected).**

| Variable | Confirmed diagnosis | Suspected diagnosis | p.value |
|---|---|---|---|
| Age at diagnosis | 75.7 q1;q3=[71.4;78.3] | 72.3 q1;q3=[62.4;78.3] | 0.3059 |
| Weight | 72 q1;q3=[62;80] | 68.8 q1;q3=[57.8;80] | 0.8936 |
| Size | 172 q1;q3=[165;177] | 170 q1;q3=[162;173] | 0.36 |
| BMI | 23.6 q1;q3=[21.5;25.8] | 24.3 q1;q3=[22.2;26.1] | 0.7411 |
| **Geographical origin** | | | |
| Cayenne | 26 (72.2%) | 8 (72.7%) | 1 |
| Eastern Guyana | 1 (2.8%) | 0 (0%) | |
| Kourou | 6 (16.7%) | 2 (18.2%) | |
| West Guiana | 3 (8.3%) | 1 (9.1%) | |
| **Case history** | | | |
| At least 1 CVRF | 17 (94.4%) | 15 (51.7%) | 0.0031 |
| Arterial hypertension | 26 (72.2%) | 6 (54.5%) | 0.2923 |
| Diabetes | 5 (13.9%) | 2 (18.2%) | 0.6593 |
| Dyslipidemia | 4 (11.1%) | 0 (0%) | 0.5597 |
| Tobacco | 0 (0%) | 0 (0%) | |
| Alcohol | 1 (2.8%) | 0 (0%) | 1 |
| **Clinical damage** | | | |
| Systolic blood pressure | 131.5 q1;q3=[117.5;142.5] | 135 q1;q3=[114;140] | 0.9732 |
| Diastolic blood pressure | 75 q1;q3=[65.2;85] | 71.5 q1;q3=[70;82.5] | 0.5009 |
| Heart rate | 77.5 q1;q3=[70;84] | 83 q1;q3=[68;96] | 0.4117 |
| *NYHA* | | | |
| I | 5 (16.7%) | 2 (20%) | 0.8441 |
| II | 13 (43.3%) | 3 (30%) | |
| III | 8 (26.7%) | 4 (40%) | |
| IV | 4 (13.3%) | 1 (10%) | |
| Previous MI | 1 (2.8%) | 0 (0%) | 1 |
| *ECG* | | | |
| Microvoltage | 10 (31.2%) | 5 (45.5%) | 0.4732 |
| Sinusal | 21 (63.6%) | 8 (80%) | 0.7286 |
| Atrial fibrillation | 8 (24.2%) | 2 (20%) | |
| Stimulation | 4 (12.1%) | 0 (0%) | |
| *Biology* | | | |
| NT pro BNP | 3345 q1;q3=[1184;7139] | 4849 q1;q3=[2476.5;6837.5] | 0.5185 |
| Hemoglobin | 13.2 q1;q3=[11.3;14.8] | 12.6 q1;q3=[10.5;13] | 0.2809 |
| Creatininemia | 106 q1;q3=[85;131.5] | 100 q1;q3=[78;158] | 0.9355 |
| Gammapathy | 7 (20.6%) | 0 (0%) | 0.5672 |
| *Echocardiographic criteria* | | | |
| LVEF | 44 q1;q3=[37;58.5] | 61.5 q1;q3=[38;65] | 0.2752 |
| LVEF<50% | 17 (54.8%) | 3 (30%) | 0.2772 |
| IVS | 18 q1;q3=[15;20] | 18 q1;q3=[18;18.8] | 0.3815 |
| IVS>12 mm | 25 (86.2%) | 6 (100%) | 1 |
| Strain | −9.4 q1;q3=[−11;-6] | −13.5 q1;q3=[−13.8;-13.2] | 0.1088 |
| *Extracardiac disorders* | 11 (30.6%) | 0 (0%) | <0.001 |
| Carpal tunnel syndrome | 18 (50%) | 0 (0%) | 0.0032 |
| Narrow lumbar canal | 5 (13.9%) | 0 (0%) | 0.3216 |

*(Continued)*

**Table 3.** (Continued)

| Variable | Confirmed diagnosis | Suspected diagnosis | p.value |
|---|---|---|---|
| Dupuytren | 1 (2.8%) | 0 (0%) | 1 |
| Peripheral neuropathy | 11 (30.6%) | 0 (0%) | 0.0457 |
| Dysautonomia | 7 (19.4%) | 0 (0%) | 0.1751 |
| Hypotension | 6 (16.7%) | 0 (0%) | 0.3121 |
| Deafness | 5 (13.9%) | 0 (0%) | 0.3216 |
| **Imaging tests** | | | |
| MRI performed | 23 (69.7%) | 6 (54.5%) | 0.4676 |
| *Bone scan* | | | |
| Hyperfixation | 25 (69.4%) | 0 (0%) | 0.0005 |
| Negative | 2 (5.6%) | 0 (0%) | |
| Not carried out | 9 (25%) | 11 (100%) | |
| Perrugini 2 | 2 (40%) | 0 (0%) | 1 |
| Perrugini 3 | 3 (60%) | 0 (0%) | |
| **Anatomopathology** | | | |
| Not done | 6 (17.1%) | 7 (63.6%) | |
| Salivary gland biopsy | 22 (62.9%) | 3 (27.3%) | |
| Endocardium | 4 (11.4%) | 0 (0%) | |
| Abdominal fat | 1 (2.9%) | 0 (0%) | |
| Kidney biopsy | 2 (5.7%) | 0 (0%) | |
| Type of biopsy unknown | 0 (0%) | 1 (9.1%) | |
| *Results* | | | |
| Positive | 25 (86.2%) | 2 (66.7%) | 0.4103 |
| Congo red positive | 23 (92%) | 2 (100%) | 1 |
| *Typing* | | | |
| TTR | 14 (66.7%) | 0 (0%) | |
| AA | 1 (4.8%) | 0 (0%) | 0.3703 |
| AL | 1 (4.8%) | 0 (0%) | |
| Unknown | 5 (23.8%) | 1 (100%) | |
| **Genetic analysis** | | | |
| Realized | 24 (66.7%) | 0 (0%) | 0.0001 |
| *Location* | | | |
| CHUM | 4 (16%) | 0 (0%) | 1 |
| Henri Mondor Center | 21 (84%) | 0 (0%) | |
| **Treatments** | | | |
| Tafamidis | 22 (61.1%) | 0 (0%) | 0.0003 |
| Diuretic | 24 (68.6%) | 8 (80%) | 0.6983 |
| Beta-blocker | 0 (0%) | 1 (10%) | 0.2222 |
| Pacemaker | 8 (22.2%) | 0 (0%) | 0.1695 |
| ICD | 9 (25%) | 0 (0%) | 0.0916 |
| Chemotherapy | 6 (16.7%) | 0 (0%) | 0.3121 |
| **Reference center** | 29 (80.6%) | 0 (0%) | <0.001 |
| **Deaths** | 8 (44.4%) | 10 (34.5%) | 0.7082 |
| *Cause of death* | | | |
| Cardiopulmonary arrest | 1 (12.5%) | 0 (0%) | |
| Cardiogenic shock | 0 (0%) | 1 (10%) | |

*(Continued)*

**Table 3.** (Continued)

| Variable | Confirmed diagnosis | Suspected diagnosis | p.value |
|---|---|---|---|
| Multivisceral failure | 2 (25%) | 1 (10%) | |
| End-stage heart failure | 2 (25%) | 3 (30%) | |
| Infection | 2 (25%) | 0 (0%) | |
| Extracardiac | 0 (0%) | 1 (10%) | |
| Unknown | 1 (12.5%) | 4 (40%) | |

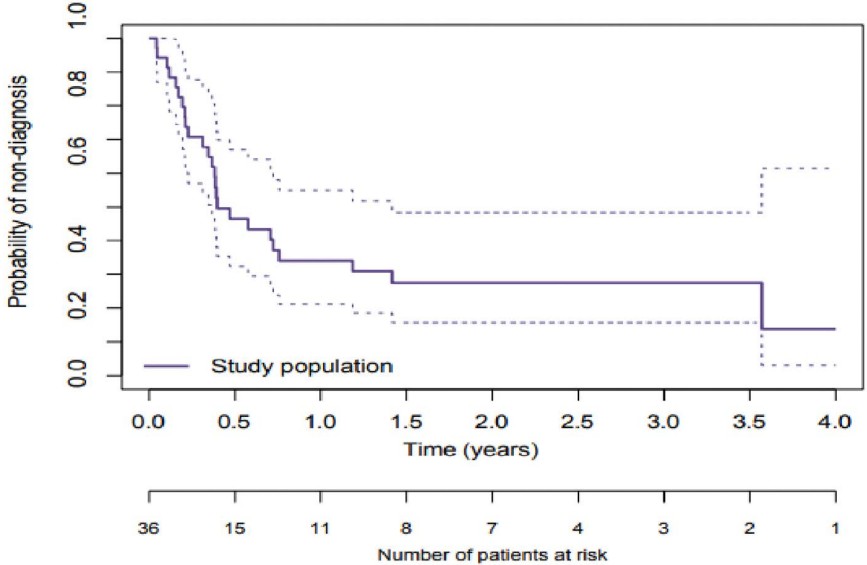

**Fig 3. Probability of non-diagnosis.**

It is difficult to find studies on all types of amyloidosis to compare with the national population in France. However, a study that overestimated the incidence of ATTR with cardiac involvement in France found a population slightly older than our ATTR patients (78.7), with a more pronounced male predominance (66.5%) [22].

One of the aims of this study was to describe the management of Guianese patients with suspected cardiac amyloidosis. The diagnosis of amyloidosis was difficult in French Guiana because there was no rapid access to bone scintigraphy, which is the main diagnostic test for non-AL amyloidosis. The results showed that this test was systematically performed only in just over half of the patients suspected of having amyloidosis during the period observed when they were referred to a reference center in mainland France or Martinique. The reasons for this lack of referral are unclear, and it would be interesting to find more in future studies. However, several hypotheses can be proposed: is the diagnosis suspected too late in patients who are too fragile to make referral to a referral center too risky? Prior to 2021, there was no specific treatment for ATTR with cardiac involvement, and therefore, less benefit from definitive diagnosis in suspected patients. Refusal of patients due to lack of knowledge about the disease and its impact on family members in the case of ATTR? Similarly, genetic analyses in ATTRm were performed in the reference centers and not in French Guiana. The lack of genetic studies can be explained by the absence of a dedicated geneticist during the study period. These barriers to early diagnosis of amyloid have been identified in other Latin American and Caribbean countries [12, 16]. Since 2023, a geneticist has been present in French Guiana. Patients can be referred to him to compensate for this lack of diagnosis.

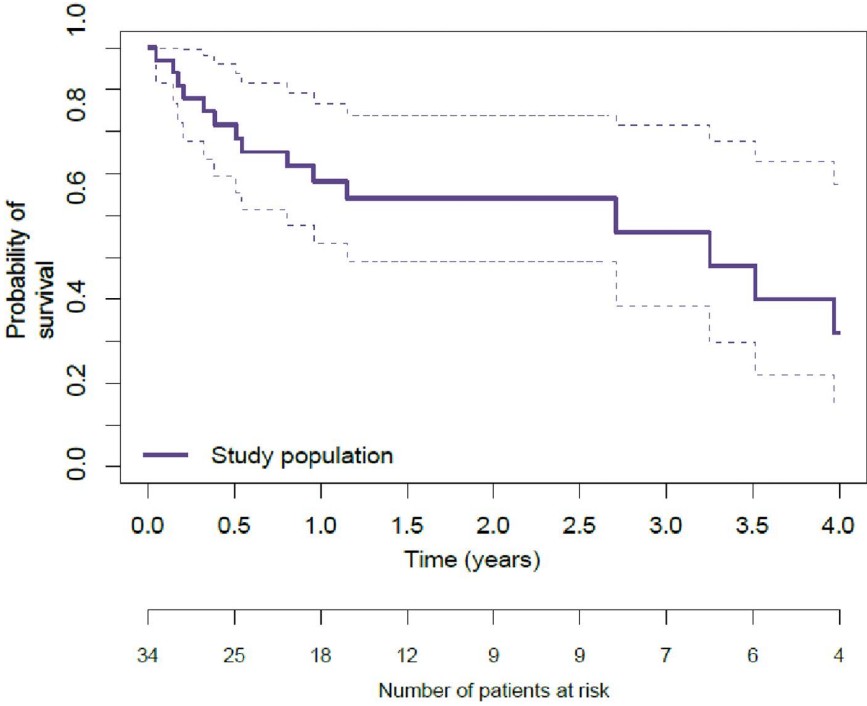

**Fig 4. Probability of survival.**

In survival analysis, the median survival time was 38 months. The 2016 study by Damy et al [23] found a median survival of 26 months in a population of patients with all types of cardiac amyloidosis (ATTRwt, ATTRm, and AL). Apart from this study, it is difficult to compare our results with those of other survival analyses because they are often performed on specific types of amyloidosis. For information, survival analyses in previous studies [22] found a median survival of 42 months in a cohort of patients with ATTR in the study by Damy et al [22], 2.7 years for ATTRwt and 0.87 years for AL amyloidosis in the study by Piney et al [24]. With a 1-year survival rate of 80%, Posadas-Martinez et al. noted that most deaths (73.1%; n = 19/26 deaths) occurred in patients with cardiac involvement [25]. Nevertheless, we note that the median survival of Guianese patients with cardiac amyloidosis was similar to that observed in the various studies cited above. Although cardiac amyloidosis is a serious disease with a guarded prognosis, our results, which seem comparable to those of the reference center, seem surprising and call for caution. In fact, the analyses focused only on patients whose diagnosis was confirmed and whose date of diagnosis was available, which represented only 34 of the 47 patients included.

In recent years, the therapeutic outlook for TTR amyloidosis has significantly improved with the emergence of several specific amyloidosis treatments. Tafamidis is currently the only treatment with marketing authorization (AMM) in France for heart disease. It is a stabilizer of the transthyretin tetramer and has been approved since 2021 [26]. Other treatments have also been studied, in particular patisiran (gene silencing therapy based on small interfering RNA (siRNA), whose action is based on the suppression of any synthesis of mutant transthyretin), which is already used in amyloid polyneuropathy and for which the APOLLO-B trial in cardiac involvement as well as HELIOS-B results for vutrisiran has been completed [27–29].

Given the recent advances in specific treatments for ATTR, it seems necessary to improve the diagnostic pathways for amyloidosis in French Guiana to achieve the earliest possible diagnosis and provide Guianese patients access to these new therapies. As part of the National Plan for Rare Diseases, a Rare Disease Coordination Platform has been set up

in French Guiana by 2020, a structure to which general practitioners and patients can refer in order to reduce diagnostic wandering and improve care for rare diseases. In this context, an amyloidosis team was established in 2022. This unit, which will bring together cardiologists, internists, general practitioners, radiologists, anatomopathologists, geneticists and the Coordination Platform for Rare Diseases in French Guiana, has developed local skills in the diagnosis of amyloidosis by focusing on cardiac MRI and biopsy, which can be performed locally. and systematize the cycle that allows each patient to receive a definitive diagnosis. It also plans to conduct research into the epidemiology and management of CA and the need for a multidisciplinary approach to improve the quality of life of patients with this rare disease.

## Limitations

Our study had several limitations. Owing to its retrospective nature, it is not certain that we were able to identify all patients who could potentially be included in the study. In addition, some clinical data were difficult to obtain, particularly from CHOG, whose archives were only available in paper form and whose location has recently changed. As the study was based on patients' pre-existing clinical data, it was only possible to assess the variables available in the database. However, these variables are absolute and widely available, which should improve the internal validity of the study. Limitations of the study include the lack of a control group of patients without cardiac amyloidosis, which makes it difficult to determine whether certain risk factors are actually more common. However, the majority of patients in the study were from Cayenne, which may have introduced geographical bias. Therefore, the obtained results cannot be extrapolated to the general population of French Guiana. We were aware of these difficulties at the beginning of the study and attempted to overcome them by obtaining data from private cardiologists in French Guiana and, in particular, from the two main reference centers to which patients were specifically referred (CHUM and Henri Mondor Hospital). This additional effort allowed us to collect the most complete data and obtain the first results for cardiac amyloidosis in French Guiana. Now that we have established a regional amyloid team, a simplified and accessible diagnostic protocol, better training for healthcare professionals, and a high level of public awareness of cardiac amyloidosis, the next step will be to establish an institutional registry of amyloidosis report.

## Conclusion

In conclusion, this study provides the first information on the diagnostic pathway for cardiac amyloidosis in French Guiana. The increasing proportion of undiagnosed patients has led us to create a French Guianese Amyloidosis Team to simplify the diagnostic pathway by focusing on cardiac MRI and biopsy, which can be performed locally. This is particularly important, as current and future therapeutic advances mean that more effective treatments are on the horizon. The limitations identified in this first study highlight the need for further research into the epidemiology and management of CA, as well as the need for a multidisciplinary approach to improve the quality of life of patients with this rare disease.

## Supporting information

**S1 Data. Data base.**
(ZIP)

## Acknowledgments

The authors would like to thank all patients who agreed to participate in this study.

## Author contributions

**Conceptualization:** Narcisse Elenga, Baptiste Desjardins, Cyrille Mathien.

**Data curation:** Baptiste Desjardins, Kénol Franck, Nathalie Deschamps, Jean-Franky Alexis, Cyrille Mathien, Franck Boteko, Grace à Dieu Yabeta, Thibaud Damy, Jocelyn Inamo.

**Formal analysis:** Baptiste Desjardins, Thibaud Damy, Jocelyn Inamo.

**Investigation:** Narcisse Elenga, Thibaud Damy, Jocelyn Inamo.

**Methodology:** Narcisse ELENGA, Baptiste Desjardins, Cyrille Mathien, Thibaud Damy.

**Project administration:** Narcisse ELENGA.

**Resources:** Cyrille Mathien.

**Software:** Jean-Franky Alexis, Cyrille Mathien.

**Supervision:** Narcisse ELENGA, Kénol Franck, Nathalie Deschamps, Jean-Franky Alexis, Cyrille Mathien, Franck Boteko, Grace à Dieu Yabeta, Thibaud Damy, Jocelyn Inamo.

**Validation:** Kénol Franck, Nathalie Deschamps, Jean-Franky Alexis, Cyrille Mathien, Franck Boteko, Grace à Dieu Yabeta, Thibaud Damy, Jocelyn Inamo.

**Writing – original draft:** Narcisse ELENGA, Baptiste Desjardins.

**Writing – review & editing:** Narcisse ELENGA, Baptiste Desjardins, Jocelyn Inamo.

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
