## [Decision Letter · Decision Letter 0]

30 Jan 2025

Dear Dr. ELENGA,

Thank you for submitting your manuscript to PLOS ONE. After careful consideration, we feel that it has merit but does not fully meet PLOS ONE’s publication criteria as it currently stands. Therefore, we invite you to submit a revised version of the manuscript that addresses the points raised during the review process.

We look forward to receiving your revised manuscript.

Kind regards,

Gbolahan Olatunji, M.D.

Academic Editor

PLOS ONE

2. In the online submission form, you indicated that your data will be submitted to a repository upon acceptance.  We strongly recommend all authors deposit their data before acceptance, as the process can be lengthy and hold up publication timelines. Please note that, though access restrictions are acceptable now, your entire minimal  dataset will need to be made freely accessible if your manuscript is accepted for publication. This policy applies to all data except where public deposition would breach compliance with the protocol approved by your research ethics board. If you are unable to adhere to our open data policy, please kindly revise your statement to explain your reasoning and we will seek the editor's input on an exemption.

Additional Editor Comments:

Please provide additional information on how you obtained a sample size of 47 patients. 

Which statistical methods did you use to make up for missing data, and how did you ensure standardization of data obtained from "private cardiologists"

Kindly improve the grammatical structure of the article

NB: Please note that revision of your article does not guarantee acceptance

Reviewers' comments:

Reviewer's Responses to Questions

**Comments to the Author**

1. Is the manuscript technically sound, and do the data support the conclusions?

Reviewer #1: Yes

Reviewer #2: No

Reviewer #3: Yes

Reviewer #4: Yes

2. Has the statistical analysis been performed appropriately and rigorously?

Reviewer #1: Yes

Reviewer #2: No

Reviewer #3: Yes

Reviewer #4: Yes

3. Have the authors made all data underlying the findings in their manuscript fully available?

Reviewer #1: Yes

Reviewer #2: Yes

Reviewer #3: Yes

Reviewer #4: Yes

4. Is the manuscript presented in an intelligible fashion and written in standard English?

Reviewer #1: Yes

Reviewer #2: No

Reviewer #3: Yes

Reviewer #4: Yes

Reviewer #1: "Unlocking Heart Health: Key Findings on Cardiac Amyloidosis in French Guiana Revealed!"

The authors performed a clinical multicenter retrospective study of Guianese patients with confirmed or suspected cardiac amyloidosis. 

The introduction is too long and should be more concise.

The introduction section should target the current study topic and related cardiac diseases. The following references might be helpful:

https://doi.org/10.1016/j.biopha.2017.04.033

https://doi.org/10.1016/j.biopha.2023.114599

Did the author consider conducting an in vitro study using appropriate cardiomyocyte cell lines to confirm the current data?

You should revise the manuscript for minor linguistic, grammatical, and style errors.

You should revise the entire manuscript to ensure proper use of abbreviations.

Did the authors check their data for normality before applying the appropriate statistical test?

The authors should explain and write the limitations of the current study.

Reviewer #2: The manuscript, titled "Unlocking Heart Health: Key Findings on Cardiac Amyloidosis in French Guiana," presents a retrospective, multicenter study evaluating the epidemiological characteristics and diagnostic pathways of cardiac amyloidosis in French Guiana. However, this study has major concerns and issues:

1. The introduction section is excessively lengthy and lacks a clear focus on the study's objectives. It includes superfluous details that detract from the main research question and reduce its overall clarity and impact.

2. The study reports a high rate of non-response or non-participation, which raises concerns about potential selection bias. This limitation significantly affects the robustness and generalizability of the findings, warranting further discussion and transparency in its potential implications.

Reviewer #3: I would like to thank the author for their contribution and manuscript. I have few comments:

1. Its better to replace "non-inclusion" to "exclusion" criteria.

2. Although cardiac amyloid is a rare entity and not commonly encountered. The sample size is very small especially over a period of 20 years of retrospective study.

3. In the conclusion section its mention "with or suspected cardiac amyloid". Are there patients with suspected cardiac amyloid which has not been confirmed?

Reviewer #4: Just 47 patients are included in the study, which is a small enough sample size to make generalizations. Furthermore, the majority of the study's patients (72.3%) are from Cayenne, which may introduce geographical bias.

Justify the small sample size and talk about if it is representative of French Guiana's overall population. Provide methods for expanding patient recruitment, if possible.

Retrospective in nature, the study uses records from private and hospital cardiologists. Since only patients who have been diagnosed or are suspected of having amyloidosis within the healthcare system are included, this could introduce selection bias.

Patients with cardiac amyloidosis are not compared to a control group of patients without the disease. It is challenging to determine whether specific risk factors are more prevalent in impacted individuals in the absence of this.

It is challenging to evaluate the clinical relevance of reported differences because the study only offers p-values without providing effect sizes or confidence intervals (CIs).

The study reports 38-month median survival, but does not compare this to other populations.

The study's results are not sufficiently compared to earlier studies on cardiac amyloidosis in Latin America, the Caribbean, or French Guiana.

**Do you want your identity to be public for this peer review?** For information about this choice, including consent withdrawal, please see our Privacy Policy

Reviewer #1: No

Reviewer #2: No

Reviewer #3: No

Reviewer #4: No

---

## [Author Response · Author response to Decision Letter 1]

17 Feb 2025

Responses to the editors and the reviewers

PONE-D-24-49901

"Unlocking Heart Health: Key Findings on Cardiac Amyloidosis in French Guiana Revealed!"

PLOS ONE

Dear Gbolahan Olatunji, M.D.

Academic Editor

PLOS ONE

Please accept the revision of our original article, « Unlocking Heart Health: Key Findings on Cardiac Amyloidosis in French Guiana Revealed ». by Elenga, Narcisse; Desjardins B, Kénol F, Deschamps N, Alexis J, Mathien C, Boteko F, Yabeta G, Damy T, Inamo J. We are grateful of the reviewer's comments and have addressed them with the point-by-point response below. We truly believe that their constructive comments have further improved our article.

Thank you for this comment. We have revised our manuscript so that it meets PLOS ONE's style requirements.

2. In the online submission form, you indicated that your data will be submitted to a repository upon acceptance. We strongly recommend all authors deposit their data before acceptance, as the process can be lengthy and hold up publication timelines. Please note that, though access restrictions are acceptable now, your entire minimal dataset will need to be made freely accessible if your manuscript is accepted for publication. This policy applies to all data except where public deposition would breach compliance with the protocol approved by your research ethics board. If you are unable to adhere to our open data policy, please kindly revise your statement to explain your reasoning and we will seek the editor's input on an exemption.

Thank you for this comment. We filed our database during this review process.

We have removed Figure 1 from our manuscript as we are unable to obtain written permission from the copyright holder to publish it.

Additional Editor Comments:

Please provide additional information on how you obtained a sample size of 47 patients.

Which statistical methods did you use to make up for missing data, and how did you ensure standardization of data obtained from "private cardiologists"

Thank you for this great comment. We have provided additional information on how we obtained a sample of 47 patients. We have also added details of the methods and provided the statistical methods used to adjust for missing data.

Kindly improve the grammatical structure of the article

We also had the whole manuscript proof-read by a recognised editor.

NB: Please note that revision of your article does not guarantee acceptance

Reviewers' comments:

Reviewer's Responses to Questions

Comments to the Author

1. Is the manuscript technically sound, and do the data support the conclusions?

Reviewer #1: Yes

Reviewer #2: No

Reviewer #3: Yes

Reviewer #4: Yes

2. Has the statistical analysis been performed appropriately and rigorously?

Reviewer #1: Yes

Reviewer #2: No

Reviewer #3: Yes

Reviewer #4: Yes

3. Have the authors made all data underlying the findings in their manuscript fully available?

Reviewer #1: Yes

Reviewer #2: Yes

Reviewer #3: Yes

Reviewer #4: Yes

4. Is the manuscript presented in an intelligible fashion and written in standard English?

Reviewer #1: Yes

Reviewer #2: No

Reviewer #3: Yes

Reviewer #4: Yes

5. Review Comments to the Author

Reviewer #1: "Unlocking Heart Health: Key Findings on Cardiac Amyloidosis in French Guiana Revealed!"

The authors performed a clinical multicenter retrospective study of Guianese patients with confirmed or suspected cardiac amyloidosis.

The introduction is too long and should be more concise.

We've rewritten the introduction and cut it in half.

The introduction section should target the current study topic and related cardiac diseases. The following references might be helpful:

https://doi.org/10.1016/j.biopha.2017.04.033

https://doi.org/10.1016/j.biopha.2023.114599

Thank you for these two references

Did the author consider conducting an in vitro study using appropriate cardiomyocyte cell lines to confirm the current data?

Great question. We are looking for a foreign team to collaborate on such a study.

You should revise the manuscript for minor linguistic, grammatical, and style errors.

Thank you. We had the whole manuscript proof-read by a recognised editor.

You should revise the entire manuscript to ensure proper use of abbreviations.

Thank done.

Did the authors check their data for normality before applying the appropriate statistical test?

Yes we did it. Thanks.

The authors should explain and write the limitations of the current study.

Yes we did it. Thanks

Reviewer #2: The manuscript, titled "Unlocking Heart Health: Key Findings on Cardiac Amyloidosis in French Guiana," presents a retrospective, multicenter study evaluating the epidemiological characteristics and diagnostic pathways of cardiac amyloidosis in French Guiana. However, this study has major concerns and issues:

1. The introduction section is excessively lengthy and lacks a clear focus on the study's objectives. It includes superfluous details that detract from the main research question and reduce its overall clarity and impact.

Thanks for your comments. We've rewritten the introduction and cut it in half, by focusing on the topic of the current study and related heart disease..

2. The study reports a high rate of non-response or non-participation, which raises concerns about potential selection bias. This limitation significantly affects the robustness and generalizability of the findings, warranting further discussion and transparency in its potential implications.

Thank you for this great comment. We have provided additional information on how we obtained our sample size. We have also added details of the methods and provided the statistical methods used to adjust for missing data. However, the limitations of the study were also discussed.

Reviewer #3: I would like to thank the author for their contribution and manuscript. I have few comments:

Thank you for comments

1. Its better to replace "non-inclusion" to "exclusion" criteria.

OK done

2. Although cardiac amyloid is a rare entity and not commonly encountered. The sample size is very small especially over a period of 20 years of retrospective study.

Thank you for this great comment. We have provided additional information on how we obtained our sample size. However, the limitations of the study were also discussed. However, 47 cases in 20 years in a current population of 300,000 seems to be a high prevalence for a rare disease.

3. In the conclusion section its mention "with or suspected cardiac amyloid". Are there patients with suspected cardiac amyloid which has not been confirmed?

Thank you for yur comment. Some changes have been made to the conclusion for clarity.

Reviewer #4: Just 47 patients are included in the study, which is a small enough sample size to make generalizations. Furthermore, the majority of the study's patients (72.3%) are from Cayenne, which may introduce geographical bias.

Thanks for your comments. The majority of patients in the study were from Cayenne, which may introduce a geographical bias. However, all patients with amyloidosis or suspected amyloidosis in French Guiana were referred to one of the hospital's cardiology departments.

Justify the small sample size and talk about if it is representative of French Guiana's overall population.

Thank you for this great comment. We have provided additional information on how we obtained our sample size. However, the limitations of the study were also discussed. However, 47 cases in 20 years in a current population of 300,000 seems to be a high prevalence for a rare disease.

Provide methods for expanding patient recruitment, if possible.

Now that we have established a regional amyloid team, a simplified and accessible diagnostic protocol, better training for healthcare professionals and a high level of public awareness of cardiac amyloidosis, the next step will be a prospective study to reduce bias and improve generalizability.

Retrospective in nature, the study uses records from private and hospital cardiologists. Since only patients who have been diagnosed or are suspected of having amyloidosis within the healthcare system are included, this could introduce selection bias.

Thanks for this great comment. We totally ahree with you. We have provided additional information on how we obtained our sample size. However, the limitations of the study were also discussed.

Patients with cardiac amyloidosis are not compared to a control group of patients without the disease. It is challenging to determine whether specific risk factors are more prevalent in impacted individuals in the absence of this.

We completely agree with you. We could do a case-control study as suggested. But first we wanted to do a cross-sectional study.

It is challenging to evaluate the clinical relevance of reported differences because the study only offers p-values without providing effect sizes or confidence intervals (CIs).

Confidence intervals have been added

The study reports 38-month median survival, but does not compare this to other populations.

We have compared survival with other populations

The study's results are not sufficiently compared to earlier studies on cardiac amyloidosis in Latin America, the Caribbean, or French Guiana.

We have compared survival with other studies, although there is little information on patients with cardiac amyloidosis in Latin American and Caribbean countries.

Yours sincerely.

---

## [Decision Letter · Decision Letter 1]

12 Mar 2025

Dear Dr. ELENGA,

Thank you for submitting your manuscript to PLOS ONE. After careful consideration, we feel that it has merit but does not fully meet PLOS ONE’s publication criteria as it currently stands. Therefore, we invite you to submit a revised version of the manuscript that addresses the points raised during the review process.

We look forward to receiving your revised manuscript.

Kind regards,

Gbolahan Olatunji, M.D.

Academic Editor

PLOS ONE

Journal Requirements:

**Additional Editor Comments:**

Please address each comments carefully, providing a detailed response to each of the comments. 

Reviewers' comments:

Reviewer's Responses to Questions

**Comments to the Author**

Reviewer #2: (No Response)

Reviewer #3: All comments have been addressed

Reviewer #4: All comments have been addressed

2. Is the manuscript technically sound, and do the data support the conclusions?

Reviewer #2: No

Reviewer #3: Yes

Reviewer #4: Yes

3. Has the statistical analysis been performed appropriately and rigorously?

Reviewer #2: No

Reviewer #3: Yes

Reviewer #4: Yes

4. Have the authors made all data underlying the findings in their manuscript fully available?

Reviewer #2: No

Reviewer #3: Yes

Reviewer #4: Yes

5. Is the manuscript presented in an intelligible fashion and written in standard English?

Reviewer #2: (No Response)

Reviewer #3: Yes

Reviewer #4: Yes

Reviewer #2: The authors have not adequately addressed the comments, and significant concerns remain regarding selection bias, which may substantially impact the final conclusions

Reviewer #3: I would like to thank the author for the revised manuscript. I would like to thank them for the changes made in relation to the provided comments and feedback. All comments were addressed with no shortcomings.

Reviewer #4: The introduction contains excessive background information and lacks focus on the study objectives. The study includes only 47 patients over 20 years, raising concerns about selection bias, with 72.3% of patients from Cayenne, leading to geographical bias. Additionally, the absence of a comparison group of patients without cardiac amyloidosis makes it difficult to determine whether specific risk factors are truly more prevalent. The handling of missing data is unclear, and some results lack effect sizes and confidence intervals. Furthermore, the Discussion section does not sufficiently compare findings with relevant studies from Latin America and the Caribbean, limiting the broader contextual relevance of the study.

**Do you want your identity to be public for this peer review?** For information about this choice, including consent withdrawal, please see our Privacy Policy

Reviewer #2: No

Reviewer #3: No

Reviewer #4: No

---

## [Author Response · Author response to Decision Letter 2]

22 Mar 2025

Responses to the editors and the reviewers R2

PONE-D-24-49901

"Unlocking Heart Health: Key Findings on Cardiac Amyloidosis in French Guiana Revealed!"

PLOS ONE

Dear Gbolahan Olatunji, M.D.

Academic Editor

PLOS ONE

Please accept the 2nd revision of our original article, « Unlocking Heart Health: Key Findings on Cardiac Amyloidosis in French Guiana Revealed ». by Elenga, Narcisse; Desjardins B, Kénol F, Deschamps N, Alexis J, Mathien C, Boteko F, Yabeta G, Damy T, Inamo J. We are grateful of the reviewer's comments and have addressed them with the point-by-point response below. We truly believe that their constructive comments have further improved our article.

Reviewer #2: The authors have not adequately addressed the comments, and significant concerns remain regarding selection bias, which may substantially impact the final conclusions

Thank you for this comment. We discussed selection bias, compared our study with other studies in Latin American and Caribbean countries, and addressed limitations.

Reviewer #3: I would like to thank the author for the revised manuscript. I would like to thank them for the changes made in relation to the provided comments and feedback. All comments were addressed with no shortcomings.

Thank you so much

Reviewer #4: The introduction contains excessive background information and lacks focus on the study objectives.

Thank you for your comments. We've rewritten the introduction, focusing on the topic of the current study and related heart diseases, as well as the objectives of the study.

The study includes only 47 patients over 20 years, raising concerns about selection bias, with 72.3% of patients from Cayenne, leading to geographical bias.

Thank you for this comment. We have discussed these biases in the manuscript.

Additionally, the absence of a comparison group of patients without cardiac amyloidosis makes it difficult to determine whether specific risk factors are truly more prevalent.

This is a major limitation of our study, which we have highlighted. Thanks.

The handling of missing data is unclear, and some results lack effect sizes and confidence intervals.

Confidence intervals have been added but were not available for all variables.

Furthermore, the Discussion section does not sufficiently compare findings with relevant studies from Latin America and the Caribbean, limiting the broader contextual relevance of the study.

Thank you for this comment. We discussed selection bias, compared our study with other studies in Latin American and Caribbean countries, and addressed limitations.

---

## [Decision Letter · Decision Letter 2]

24 Apr 2025

Dear Dr. ELENGA,,

Thank you for submitting your manuscript to PLOS ONE. After careful consideration, we feel that it has merit but does not fully meet PLOS ONE’s publication criteria as it currently stands. Therefore, we invite you to submit a revised version of the manuscript that addresses the points raised during the review process.

**ACADEMIC EDITOR:**
** **

Please submit your revised manuscript by Jun 08 2025 11:59PM. If you will need more time than this to complete your revisions, please reply to this message or contact the journal office at plosone@plos.org . A rebuttal letter that responds to each point raised by the academic editor and reviewer(s). You should upload this letter as a separate file labeled 'Response to Reviewers.'A marked-up copy of your manuscript that highlights changes made to the original version. You should upload this as a separate file labeled 'Revised Manuscript with Track Changes.'An unmarked version of your revised paper without tracked changes. You should upload this as a separate file labeled 'Manuscript.'

If applicable, we recommend that you deposit your laboratory protocols in protocols.io to enhance the reproducibility of your results. Protocols.io assigns your protocol its own identifier (DOI) so that it can be cited independently in the future. For instructions, see https://journals.plos.org/plosone/s/submission-guidelines#loc-laboratory-protocols. Additionally, PLOS ONE offers an option for publishing peer-reviewed Lab Protocol articles, which describe protocols hosted on protocols.io. Read more information on sharing protocols at https://plos.org/protocols?utm_medium=editorial-email&utm_source=authorletters&utm_campaign=protocols .

We look forward to receiving your revised manuscript.

Kind regards,

Gbolahan Olatunji, M.D.

Academic Editor

PLOS ONE

Journal Requirements:

Reviewers' comments:

Reviewer's Responses to Questions

**Comments to the Author**

Reviewer #5: (No Response)

Reviewer #6: All comments have been addressed

2. Is the manuscript technically sound, and do the data support the conclusions?

Reviewer #5: Yes

Reviewer #6: Yes

3. Has the statistical analysis been performed appropriately and rigorously?

Reviewer #5: Yes

Reviewer #6: Yes

4. Have the authors made all data underlying the findings in their manuscript fully available?

Reviewer #5: Yes

Reviewer #6: Yes

5. Is the manuscript presented in an intelligible fashion and written in standard English?

Reviewer #5: Yes

Reviewer #6: Yes

Reviewer #5: I commend the authors’ work in revealing key findings on cardiac amyloidosis in French Guiana and their extensive revisions and responses to the previous reviewers’ comments.

I have a few comments:

- Some sentences/facts were not cited in the introduction section - please review and cite appropriately

- Authors should be consistent with the use of exclusion criteria vs. non-inclusion“ criteria—"exclusion criteria” is preferred, as initially noted by Reviewer 3. See pg 6/47 of “Amylose.GuyaneRevisedCLEAN(2)” and adjust

Reviewer #6:  Thank you for your paper.

The data was analyzed intelligently and clearly for interpretation.

I suggest that a heading of "LIMITATIONS" be included before the paragraph that outlines the study's limitations.

If you choose “no,” your identity will remain anonymous, but your review may still be made public.

**Do you want your identity to be public for this peer review?** For information about this choice, including consent withdrawal, please see our Privacy Policy

Reviewer #5: No

Reviewer #6: No

[NOTE: If reviewer comments were submitted as an attachment file, they will be attached to this email and accessible via the submission site. Please log into your account, locate the manuscript record, and check for the action link "View Attachments." If this link does not appear, there are no attachment files.

While revising your submission, please upload your figure files to the Preflight Analysis and Conversion Engine (PACE) digital diagnostic tool, https://pacev2.apexcovantage.com/ . PACE helps ensure that figures meet PLOS requirements. To use PACE, you must first register as a user. Registration is free. Then, log in and navigate to the UPLOAD tab, where you will find detailed instructions on how to use the tool. If you encounter any issues or have any questions when using PACE, please email PLOS at figures@plos.org

---

## [Author Response · Author response to Decision Letter 3]

27 Apr 2025

Responses to the editors and the reviewers

PONE-D-24-49901

" A Comprehensive Evaluation of Cardiac Amyloidosis Epidemiology and Diagnostics in French Guiana"

PLOS ONE

Dear Gbolahan Olatunji, M.D.

Academic Editor

PLOS ONE

Please accept the 3rd revision of our original article, « A Comprehensive Evaluation of Cardiac Amyloidosis Epidemiology and Diagnostics in French Guiana». by Elenga, Narcisse; Desjardins B, Kénol F, Deschamps N, Alexis J, Mathien C, Boteko F, Yabeta G, Damy T, Inamo J. We are grateful of the reviewer's comments and have addressed them with the point-by-point response below. We truly believe that their constructive comments have further improved our article.

ACADEMIC EDITOR:

Kindly include what type of article it is in the title and consider tweaking the title a little to be more scientific.

Thank you for this comment.

The new title is : « A Comprehensive Evaluation of Cardiac Amyloidosis Epidemiology and Diagnostics in French Guiana »

Journal Requirements:

We have reviewed and corrected all references

Reviewer #5: I commend the authors’ work in revealing key findings on cardiac amyloidosis in French Guiana and their extensive revisions and responses to the previous reviewers’ comments.

I have a few comments:

- Some sentences/facts were not cited in the introduction section - please review and cite appropriately

Thank you for this comment. We have added some sentences to describe Cardiad Amylioidosis

- Authors should be consistent with the use of exclusion criteria vs. non-inclusion“ criteria—"exclusion criteria” is preferred, as initially noted by Reviewer 3. See pg 6/47 of “Amylose.GuyaneRevisedCLEAN(2)” and adjust

OK corrected , thanks

Reviewer #6: Thank you for your paper.

The data was analyzed intelligently and clearly for interpretation.

I suggest that a heading of "LIMITATIONS" be included before the paragraph that outlines the study's limitations.

We have added a title « Limitations » as suggested.

---

## [Decision Letter · Decision Letter 3]

5 May 2025

"

A Comprehensive Evaluation of Cardiac Amyloidosis Epidemiology and Diagnostics in French Guiana

"

PONE-D-24-49901R3

Dear Dr. Elenga,

We’re pleased to inform you that your manuscript has been judged scientifically suitable for publication and will be formally accepted for publication once it meets all outstanding technical requirements.

Kind regards,

Gbolahan Deji Olatunji, M.D.

Academic Editor

PLOS ONE

Additional Editor Comments (optional):

Reviewers' comments:

Reviewer's Responses to Questions

**Comments to the Author**

Reviewer #7: All comments have been addressed

2. Is the manuscript technically sound, and do the data support the conclusions?

Reviewer #7: Yes

3. Has the statistical analysis been performed appropriately and rigorously?

Reviewer #7: Yes

4. Have the authors made all data underlying the findings in their manuscript fully available?

Reviewer #7: Yes

5. Is the manuscript presented in an intelligible fashion and written in standard English?

Reviewer #7: Yes

Reviewer #7: I have carefully reviewed the revised manuscript and I am pleased to confirm that all previously raised comments and concerns have been adequately addressed.

**Do you want your identity to be public for this peer review?** For information about this choice, including consent withdrawal, please see our Privacy Policy

Reviewer #7: **Yes: ** Oluwadamilola Bolanle

---

## [Editor Report · Acceptance letter]

PONE-D-24-49901R3

PLOS ONE

Dear Dr. ELENGA,

I'm pleased to inform you that your manuscript has been deemed suitable for publication in PLOS ONE. Congratulations! Your manuscript is now being handed over to our production team.

Kind regards,

on behalf of

Dr. Gbolahan Deji Olatunji

Academic Editor

PLOS ONE